# Assessing Inequity in Green Space Exposure toward a “15-Minute City” in Zhengzhou, China: Using Deep Learning and Urban Big Data

**DOI:** 10.3390/ijerph19105798

**Published:** 2022-05-10

**Authors:** Jingjing Luo, Shiyan Zhai, Genxin Song, Xinxin He, Hongquan Song, Jing Chen, Huan Liu, Yuke Feng

**Affiliations:** 1Key Laboratory of Geospatial Technology for the Middle and Lower Yellow River Regions (Henan University), Ministry of Education, Kaifeng 475004, China; ljj18238220679@126.com (J.L.); gxsong@henu.edu.cn (G.S.); 104753190133@henu.edu.cn (X.H.); hqsong@henu.edu.cn (H.S.); chenjing9811@163.com (J.C.); liuhuan@vip.henu.edu.cn (H.L.); fyk2017901110@126.com (Y.F.); 2College of Geography and Environmental Science, Henan University, Kaifeng 475004, China

**Keywords:** green space exposure, inequity, street view images, deep learning

## Abstract

Green space exposure is considered an important aspect of a livable environment and human well-being. It is often regarded as an indicator of social justice. However, due to the difficulties in obtaining green space exposure data from a ground-based view, an effective evaluation of the green space exposure inequity at the community level remains challenging. In this study, we presented a green space exposure inequity assessment framework, integrating the Green View Index (GVI), deep learning, spatial statistical analysis methods, and urban rental price big data to analyze green space exposure inequity at the community level toward a “15-minute city” in Zhengzhou, China. The results showed that green space exposure inequality is evident among residential communities. The areas in the old city were with relatively high GVI and the new city districts were with relatively low GVI. Moreover, a spatially uneven association was observed between the degree of green space exposure and housing prices. Especially, the wealthier communities in the new city districts benefit from low green space, compared to disadvantaged communities in the old city. The findings provide valuable insights for policy and planning to effectively implement greening strategies and eliminate environmental inequality in urban areas.

## 1. Introduction

Over the past century, environmental justice has been a global issue [1]. The concept of environmental justice is based on the universal principle that all people have the right to be protected from specific environmental problems (e.g., pollution, noise) and to access the same services (e.g., green space, transport) [2]. Environmental inequality focuses on the broader level of intersection between environmental quality and social class. Environmental inequality addresses structural issues such as social inequality (the unequal distribution of power and resources in society) and environmental burden [3]. Most research on environmental justice is mainly focused on pollution exposure [4,5], green exposure [6], and green space accessibility [7,8]. In recent years, green space inequity, as part of these aspects, has attracted particular attention [9,10]. In terms of access to green space, environmental equality refers to residents’ fair access to green space, not affected by other residents’ factors. Environmental inequality indicates the uneven distribution of green space among residents [10].

The COVID-19 outbreak has had a significant impact on the civilian economy. The past concept of urban development of “material-oriented” and “seeing things but not seeing people” has been reconsidered and turned into the people-oriented concept of urban development [11]. How to build more resilient and sustainable cities has attracted worldwide attention. In 2018, the new urban Residential Area Planning and Design Standard released by the Ministry of Housing and Urban-Rural Development, the “15-min city “, “10-min city” and “5-min city” are taken as the core objects of residential area planning and facility configuration. The reasonable planning of street green space has undoubtedly become an important part of building a 15-min high-quality community city.

Urban street greening is an important component in the urban landscape, supporting the ecological environment [12]. Urban street greening also plays an important role in making neighborhoods more attractive and walkable [13]. With the construction of livable cities, the government has spared no effort to improve the walkability of cities, and green space has increasingly become an important standard to measure the livable environment of cities [7]. Several studies have shown that green space may reduce people’s exposure to air pollution [14] by promoting carbon sequestration and oxygen production, absorbing air pollutants, and mitigating the urban heat island’s effect. In addition, green spaces can restore people’s attention and improve mental and physical health by reducing stress [15,16]. In fact, people are becoming increasingly aware of the benefits of green space in creating a livable urban environment. However, many previous studies showed that urban green space is not equally distributed in cities, thus urban residents and communities may not be able to equally benefit from green space [17,18,19,20].

A large number of studies on the topic of environmental inequality have been conducted in developed countries [21,22,23]. These studies reported that urban green exposure varies according to socioeconomic status (SES). Generally, neighborhoods with higher SES usually own greater financial resources, cultural and social capital, and political influences to maintain and improve green space [24], while low-income communities and communities of color have limited green space investments [25,26]. In addition, vulnerable groups cannot equally enjoy the services of green space [19,27]. For example, Li et al. [21] conducted a quantitative analysis of the spatial distribution of different types of urban green space in Hartford, Connecticut, using green indicators calculated based on multi-source spatial data sets, and found that higher-income neighborhoods in Hartford, Connecticut tend to have more green streets compared to low-income neighborhoods. Astell-Burt et al. [28] used a negative binomial and Logit regression model to investigate the relationship between green space availability and socio-economic environment, and found that green space availability was substantively lower with a higher percentage of low-income residents in Australia. Pham et al. [29] found differences in vegetation distribution in Montreal by extracting satellite images of Montreal from constant high resolution, which was unfavorable to low-income people. In China, studies in Shenzhen [16], Shanghai [30], and Guangzhou [31] revealed a direct relationship between SES and inequity of green space provision; people with higher SES enjoy more green space resources, and in affluent areas, the amount of and accessibility to public green space is better.

Previous research methods on green inequalities mainly focus on the green space in or around residential areas and take the neighborhood as the unit of analysis from the perspective of a 2D view [22,29,32]. Based on satellite remote sensing image data, the green space exposure is mainly measured by the coverage rate, area, and quantity of green space [29,33,34]. The common indicators for calculating green space included the NDVI and Leaf Area Index [16,20,35,36]. The mean NDVI method is convenient for horizontal and vertical comparison, which is intuitive and easy to calculate. However, there may be several limitations to adopting 2D remote sensing images to access green space exposure data. Firstly, it is inadequate for monitoring the personal exposure degree and cannot accurately assess personal daily exposure to the natural environment. Secondly, the low resolution of remote sensing images (e.g., 30 m) may result in biased estimates of indicators, consequently resulting in inaccurate results in the case of street-level monitoring [37,38]. While high-resolution images provide a great tool for depicting green Spaces at a fine level, they are not always available and are expensive to collect. Thirdly, the monitoring of the green space by satellite images is based on a bird’s-eye view, rather than a ground-based view [39]. They do not take into account the green-covered side view, which cannot capture the street plan and vertical section of urban greening [24]. In addition, there is not a high consistency between objectively derived green from remote sensing images and perceived green by humans [40], which is greatly affected by distance threshold or regional division and does not consider road network and actual availability.

The panoramic image analysis method evaluates the characteristics of street greening from the perspective of the human eye. Perceived green is directly related to the benefits provided by street greening. Some companies such as Google Maps, Baidu Maps, Tencent Maps, etc., provide users with 360-degree panoramic images of cities, streets, or other environments for free. Therefore, a few studies have used street view image data to extract green spatial indexes such as the green visual rate from the human eye perspective to measure green space exposure [29,41]. Some scholars have examined the distribution of street greening, including street trees, lawns, and other green spaces along the street [20,21,23]. Residential street greening, as a component of urban green space, makes an important contribution to the attractiveness and walkability of streets [12]. This represents what people really see from the ground. Yang et al. [42] developed the GVI to assess the visibility of the surrounding urban forest using color images as a proxy for pedestrians’ perception of green space. Li et al. [43] developed a new method based on Google Street View (GSV) to assess the spatial distribution of street greening. Recent developments in machine learning methods combined with online map data allow people to combine the sentiment of residents towards green spaces from social media (Flickr) with ground objects (e.g., trees and grass) from interactive panoramas (e.g., street view images (SVIs)) to better capture indicators of the quality of urban green spaces. Therefore, SVIs have been used to assess the amount of green space at eye level [44,45]. Street View is an interactive digital map that provides panoramic city street maps for users with 360°, as a representative of the urban landscape. It creates a seamless tour of the city’s streets that can feel immersive. This is very similar to what one might see exploring a city by car, bike, or foot [43], and it provides a large number of image resources for urban visual greening research. It has the advantages of high reachability, high resolution, and wide coverage. Unlike remote sensing techniques, the GSV-based method quantifies the green space at the street level, resulting in more accurate results. For example, Yin et al. [46] proposed an automated pedestrian detection and counting tool based on GSV images and machine learning algorithms to help local planners in the walkability improvement in several cities in the United States (USA). Seiferling et al. [47] measured and evaluated the number of tree canopies perceived by pedestrians along roads in Boston and New York in the USA using a large number of SVI datasets. In addition, Liu et al. [48] proposed a useful tool for measuring physical fitness in the streets of Beijing in China, using GSV images and machine learning algorithms. Others such as Helbich et al. [44] have studied the relationship between geriatric depression and street view greening obtained by deep learning and street view data using correlation analysis, to ensure effective street greening planning that supports human health. However, there are still two shortcomings in the current research. First, the green space inequity was analyzed based on 2D remote sensing images, but research based on the ground-based view is rare. Second, how to calculate the green space exposure distribution at the community level and assess green space inequity from a 15-min city perspective is still not comprehensively understood.

To fill the aforementioned research gap and better inform urban planners and policymakers in order to promote urban green equity and sustainable urban development, this study selected Zhengzhou as the case study to provide a framework for green space exposure inequity assessment, combining GVI, deep learning, spatial statistical analysis methods, and urban rental price big data toward a “15-min city” in China. This will enable the analysis of the green space exposure patterns based on the ground-based view and the assessment of its spatial inequity. Our specific objectives are: (1) To quantify the concept of “visual contact with green space”, using street view images to complement the traditional variables associated with green space; (2) to use location entropy to analyze the inequity of urban green space exposure; and (3) to use bivariate Moran I to evaluate the relationship between green space exposure and community SES. This study can provide a reference for managing urban green space, thus ensuring the sustainable development of the study area.

## 2. Materials and Methods

### 2.1. Study Area

Zhengzhou is located in the eastern part of China (Figure 1). It is the central city of the central Plains Economic Zone, China’s important railway, aviation, highway, and another major hub city. It is a developing megalopolis in central China with a population of more than 22 million. However, compared with other cities, the actual level of economic development and urbanization is low. Since the beginning of the new century, the urban functions have been diversified and the core functions of the central city have been enhanced. Zhengzhou has gradually formed a double-core complex spatial structure consisting of the original urban core and the new development zone of eastern Zhengzhou, including 9 districts, 1 county, and 5 county-level cities. The research area of this paper is limited to five districts of Zhengzhou. The built-up area of the central city is about 1006 km^2^. Based on the analysis of residential areas, 499 spatial elements were studied.

In recent years, the scale of Zhengzhou has expanded rapidly. In the latest stage of urban expansion, Zhengzhou has formed a concentric ring structure of the first, second, third, and fourth rings, and residential areas have gradually shifted from the core area to the outside [8]. Zhengzhou is one of the fastest urbanization cities, with a typical regional urban expansion. In 2017, Zhengzhou was classified as one of China’s new first-tier cities [49]. Due to the rapid development of urbanization, the expansion and population density of cities has been accelerated, which threatens the urban green space and seriously damages the ecological environment. In Zhengzhou, several key tasks and major projects are planned for enhancing green and low-carbon transformation.

### 2.2. Data Sources

#### 2.2.1. Street View Images Data

The green space refers to areas of green vegetation (e.g., grass, trees, and shrubs) in the image, measured by the GVI. The GVI is the ratio of the total green area of four images taken at an intersection to the total area of the four images. Street view images mainly measure street vegetation at eye level and the green space between neighborhoods in a three-dimensional space from a human perspective [44]. In this study, the semantic segmentation tool of visual images based on a deep learning full convolutional network (FCN) was used to obtain the scene classification results of the street view panorama of each sample point, and then the GVI of each sample point was calculated according to the classification results. In this study, each street in downtown Zhengzhou was segmented with an interval of 200 m, and the segmented points were considered as sample points to obtain street view images of each sample point at 0°, 90°, 180°, and 270° from the public API interface of Baidu Maps (https://map.baidu.com/). Street view images were collected from Baidu Maps on 23 July 2020. Some images were obtained during non-green seasons. We visually checked the vegetation conditions in each image and deleted those sites with images captured during non-green seasons. we locally adjusted the interval of sampling points (50 m) to obtain more images and correct them. This resulted in 7994 sampling points and street view images. The amount of street view green space for each sampling point was determined as the proportion of the average green space quality of the four images taken from different main directions. In previous studies, it was found that the distance factor was the main factor affecting residents’ travel. For the purpose of walking and exercising, the maximum psychological endurance time of travelers was usually 30 min [50,51]. Therefore, 30 min walking distance was selected as the maximum buffer radius in this study. Based on the walking speed per minute (72 m/min) of residents in “15-min cities”, we calculated the walking buffer distances of 5 min (360 m), 10 min (720 m), 15 min (1080 m), 30 min (2160 m). Buffer zones with 320 m, 720 m, 1080 m, and 2160 m distance radii were established according to the community boundary geographical location of the residential area. For each community cell, the green space exposure quality was calculated using the average GVI value of all sampling points within the polygon buffer boundary of the cell, which is considered as the GVI of the cell under different buffer radii.

#### 2.2.2. Housing Price Data

In the context of the real estate market boom, housing prices were used to indicate the SES of residents [7]. Green space is increasingly considered an environmental advantage of luxury residential areas over disadvantaged neighborhoods [52]. Therefore, the housing price can reflect the affordability of the community residents to obtain a green space. In this study, the housing (rental) price data from the secondary site Anjuke (https://zhengzhou.anjuke.com), accessed on 20 December 2020, which was used to assess the neighborhood SES of residents. Anjuke provides users with housing information, including second-hand housing, new housing, rentals, and so on. There is a complete residential community attribute on the Anjuke website, including the type of community address, property, price, building area, property rights, the fixed number of years, green rate, etc. We used Python code to extract residential community attributes from the real estate section of Anjuke. The data contains 499 residential communities within the central city of Zhengzhou.

### 2.3. Data Analysis

#### 2.3.1. Framework Design

The green space exposure inequity assessment framework combining GVI, deep learning, spatial statistical analysis methods, and urban rental price big data toward a “15-min city” is developed and presented in Figure 2. First, we obtained street view images (0°, 90°, 180°, 270°) from the road network from Baidu Maps. Second, the semantic segmentation of street view images was carried out using a deep learning method. We calculated the green space exposure at the community scale by use of the buffer (5-min, 10-min, 15-min, 30-min) method. Third, we adopted the Python program (python 3.7) to obtain the urban rental big data at the community level from Anjuke, which was used to indicate each community’s socioeconomic status. Finally, we assessed the green space exposure inequity using spatial statistical analysis, such as bivariate Moran’s I and the location entropy method. Figure 2 shows the framework of this study.

#### 2.3.2. Image Segmentation Based on Machine Learning

In order to extract street-view green objects (e.g., grass, trees, and shrubs), semantic image segmentation through a fully convolutional neural network (FCN-8s) was used [44,53]. This method was proposed by Long et al. [53] using classical classification networks. Based on pixel comparison and manual segmentation [54], the accuracy of the FCN-8s was reasonably high. Essentially, FCN-8s consists of a number of processing layers that connect the input layer and the output layer (semantically segmented images) in order to learn the different levels of the abstraction of data. For the input street view images, the convolution layer extracts the features, and the pooling layer compresses the data to learn the advanced feature map while reducing the spatial dimension of the feature map. Figure 3 is an example diagram of semantic segmentation.

Yang et al. [42] proposed the GVI to assess the visibility of urban forests. Street landscape greening at each sampling point was defined as the ratio of the number of green pixels in each image in four directions (east, west, south, and north) to the total number of pixels in each image in the four directions. In order to measure the streetscape greening, based on the edge of the residential area, the average image specific green space of each community was calculated according to the buffers of 5 min (360 m), 10 min (720 m), 15 min (1080 m), and 30 min (2160 m), using the following formula:(1)$$\mathrm{GVI}=\frac{\sum_{n=1}^{4}Area_{g\_n}}{\sum_{n=1}^{4}Area_{t\_n}}$$
where $Area_{g\_n}$ is the number of green pixels in an image taken in the *n*th of the four directions (east, west, south, and north) of a sample point; $Area_{t\_n}$ is the total number of pixels in an image taken at the *n*th direction of the sampling point.

#### 2.3.3. Location Entropy

The location entropy method can analyze the spatial social equity performance [55]. In this study, the location entropy analysis method was used to assess the spatial equity distribution pattern of urban public green space resources by housing price unit and analyze the distribution of urban public green space resources according to different social income groups. The location entropy index of each space unit is the ratio between the public green space by income group within the spatial unit and the public green space by unit income group within the entire study area. The calculation formula is:(2)$${\mathrm{LQ}}_{i\;}=({\mathrm{GVI}}_{i}/{\mathrm{price}}_{i})/(\mathrm{GVI}/\mathrm{price})$$

${\mathrm{LQ}}_{i}$ is the location entropy index of community *i*, GVI*_i_* is the average occupancy of green space of community *i*, ${\mathrm{price}}_{i}$ is the average housing price of community *i*, and GVI is the sum of the mean green space of the study area. Price is the sum of average rental prices in the study area.

#### 2.3.4. Spatial Statistical Analysis

Spatial autocorrelation is the most suitable method for systematically determining the unequal spatial pattern of urban green space exposure. In this paper, global bivariate Moran’s I and local bivariate Moran’s I [56] were used both to assess the spatial autocorrelation between housing price and green exposure. The formula gives the basic principles of global and local bivariate Moran statistics:(3)$$I_{P,A}=\frac{N\sum_{i}^{N}\sum_{j\neq i}^{N}W_{ij}Z_{i}^{P}Z_{j}^{A}}{\left(N-1\right)\sum_{i}^{N}\sum_{j\neq i}^{N}W_{ij}}$$
(4)$$I{’}_{P,A}\;=\sum_{j=1}^{N}W_{ij}Z_{j}^{A}$$

I_P,A_ and I’_P,A_ are global bivariate Moran’s I and local bivariate Moran’s I, respectively; *N* is the total number of cells; $Z_{i}^{P}$ is the standardized value of housing price in the *i*th community, $Z_{j}^{A}$ is the standardized value of green exposure in *j*th cell; $W_{ij}$ is the spatial weight matrix of *i*; and *j* is used to determine the correlation between the *i*th and *j*th. In this study, GeoDa was used to calculate the binary Moran’s I.

The Moran’s I values vary between −1 and 1. For bivariate Moran’s I with statistical significance, a positive value indicates spatial clustering (spatial positive correlation), while a negative value implies spatial dispersion (spatial negative correlation) [57]. Moran’s I > 0 reflects positive spatial correlation, and the larger the value is, the more obvious the spatial correlation is; Moran’s I < 0 means negative spatial correlation, and the smaller the value is, the greater the spatial difference is, whereas, a value of 0 for Moran’s I implies random space.

Global Moran’s I reflects the overall spatial autocorrelation level of neighborhood green space [20]. Therefore, to determine the spatial autocorrelation coefficient between housing price and green exposure in each community, local bivariate Moran’s I was used for detection.

Based on the spatial autocorrelation concept, cluster mapping by local bivariate Moran’s I is considered to be an effective method for detecting environmental exposure inequity hot spots or clustering regions. The cluster graph obtained by local bivariate Moran’s I can be used to identify four types of spatial correlations between housing price and GVI at the community level: Low-Low (low rental prices surrounded by low GVI), Low-High (low rental prices surrounded by high GVI), High-High (high rental prices surrounded by high GVI), and High-Low (high rental prices surrounded by low GVI).

## 3. Results

### 3.1. Spatial Inequity of Green Space Exposure

A location entropy of the community greater than 1 indicates that the urban green space services enjoyed by the population with unit income in the region are higher than the overall average level. While a location entropy of the community less than 1 indicates that the urban green space services enjoyed by the population with unit income in the region are lower than the overall average level.

Table 1 shows the location entropy value of each spatial unit within the buffer zone of the residential area, which is divided into seven levels for a certain income level and the average occupancy of urban public green space. Figure 4 shows the spatial distribution pattern of the location entropy values of each community. In order to analyze the unfairness of urban green space more accurately, we only focused on the results where the location entropy was less than 0.5 and greater than 2. First of all, the areas with very low location entropy (a location entropy value lower than 0.5; that is, the urban public green space service enjoyed by each income level is less than half of the average level) number 55 in the buffer zone of 360 m, accounting for 11.02% of the total number of cells; 42 in the buffer zone of 720 m, accounting for 8.42% of the total number of cells; and 41 in the buffer zone of 1080 m, accounting for 8.22% of the total number of cells. Moreover, there are 31 cells within the 2160 m buffer zone, accounting for 6.21% of the total number of cells. Secondly, for areas with high location entropy (location entropy value is higher than 2; that is, the urban public green space service enjoyed by each income level is two times higher than the average level), the 360 m buffer contains 41 cells, accounting for 8.22% of the total cell number; the 720 m buffer contains 41 cells, accounting for 8.22% of the total cell number; the 1080 m buffer contains 37 cells, accounting for 7.41% of the total cell number; and the 2160 m buffer contains 28 cells, accounting for 5.61% of total cell number.

As seen in Figure 4, the location entropy index showed a gradual decrease in the circle structure, and the number of communities with a higher location entropy index decreased from the inner ring to the periphery of the city, forming a circle layer structure. It can be highlighted that there is an inequality mismatch in the allocation of green public space resources in the residential areas in the study area. The communities with a lower location entropy index are mainly distributed along the outer ring. The communities with a lower location entropy index were mainly distributed in the periphery of the city, including the east of Jinshui District, the east of Erqi District, and the northeast Guancheng Minority District. The communities with a higher location entropy index were mainly distributed in the city center, including the west of Jinshui District, the east of Zhongyuan District, the north of Erqi District, and the northwest of Guancheng Minority District.

### 3.2. The Association between Green Space Exposure and Rental Prices

Figure 5 shows the spatial distribution of rental prices, relatively low in the inner city, such as the southwest of Jinshui District, the northeast of Erqi District, and the northwest of Guancheng Minority District. Rental prices in the new development zone of eastern Zhengzhou are relatively high. In this study, first, we adopted the global bivariate Moran’s I to measure the correlation between green space exposure and socioeconomic status. Table 2 indicates that there exists a negative spatial correlation between housing price and green space exposure. That is, a community with a higher housing price generally enjoys low green space exposure. Next, the bivariate LISA was applied for locally examining the association between socioeconomic conditions and green space exposure.

As shown in Figure 6, local Moran’s I revealed uneven spatial distribution between green space and different income groups, especially in High-Low (high housing price and low GVI) and Low-High (low housing price and high GVI) types.

As shown in Figure 6a, the High-Low type (high housing price and low GVI), comprised of 105 residential areas, is mainly distributed along the east new city district. Low-High type (low housing price and high GVI) are comprised of 124 residential areas concentrated in the north part of Erqi District and the southwest part of Jinshui District. On the other hand, the Low-Low type (low housing price and low GVI) comprises 85 residential areas located in the eastern part of Erqi District and the western part of Guancheng Minority District, while the High-High type (high housing price and high GVI) consists of 74 residential areas, mainly concentrated in the western part of Jinshui District.

As shown in Figure 6b, regarding the 720 m buffer zone considered, the results showed that the High-Low type (high housing price and low GVI) comprises 104 communities, mainly concentrated in the north part of Guancheng Minority District and the new development zone of eastern Zhengzhou, while the Low-High type (low housing price and high GVI) comprises 130 residential areas, mainly distributed in the southwest part of Jinshui District and in the north part of Erqi District. Moreover, the Low-Low type (low housing price and low GVI) comprises 88 residential areas, mainly concentrated in the east of Erqi District and the western part of Guancheng Minority District, whereas the High-High type (high housing price and high GVI) comprises 75 residential areas, mainly concentrated in the western part of Jinshui District.

As shown in Figure 6c, regarding the 1080 m buffer zone considered, the High-Low type (high housing price and low GVI) consists of 102 communities located in the northeast part of Guancheng Minority and the east new city district, while the Low-High type (low housing price and high GVI) consists of 133 residential areas, mainly distributed in the north of Erqi District and the southwest part of Jinshui District. Additionally, the Low-Low type (low housing price and low GVI) consists of 87 residential areas, mainly concentrated in the east of Erqi District and the western part of Guancheng Minority District, whereas the High-High type (high housing price and high GVI) consists of 73 residential areas, mainly concentrated in the western part of Jinshui District.

On the other hand, regarding the 2160 m (Figure 6d) buffer zone considered, the High-Low type (high housing price and low GVI) consists of 104 communities distributed in the northeast part of Guancheng Minority and the east new city district, while the Low-High type (low housing price and high GVI) consists of 141 residential areas, mainly distributed in the Zhongyuan District, the southwest part of Jinshui District, and the north part of Erqi District. Moreover, the Low-Low type (low housing price and low GVI) comprises 85 residential areas, mainly concentrated in the east of Erqi District and the west part of Guancheng Minority District, whereas the High-High type (high housing price and high GVI) comprises 83 residential areas, mainly concentrated in Jinshui District and Zhongyuan District.

As shown in Figure 6, the High-Low type (high housing price and low GVI) are mainly distributed along the new development zone of eastern Zhengzhou. This may be because the eastern part of Zhengzhou is a new development zone with large development space, so the housing prices are relatively high. However, various facilities are not perfect and the green environment is under planning, so the GVI value is relatively low. Low-High type (low housing price and high GVI) residential areas are mainly within the inner ring road, including the old urban area. This may be due to the shabby houses and traffic congestion in the old city, and the housing prices are relatively low. However, the old urban area, usually with high street density and good greening, is more suitable for residents to walk for leisure. The GVI is relatively high.

## 4. Discussion

This study took Zhengzhou as the research object, combined with GVI, deep learning, spatial statistical analysis methods, and the big data of urban rent prices, and constructed an evaluation framework of the unfairness of urban green space exposure, and analyzed the unfairness of green space exposure at the community level of a “15-min city” in Zhengzhou. The results show that the exposure inequality of green space in residential areas is obvious, and the index of green space in old urban areas is higher than that in the new development zone of eastern Zhengzhou. In particular, the more affluent communities in the new development zone have a lower green space compared with the disadvantaged communities in the old urban areas.

In our study, we found that there existed a gradual decrease in circle structure from the city center to the peri-urban areas, indicating that the community residents in the central urban area of Zhengzhou may better benefit from street view greening when compared to the suburbs, which is consistent with the results reported in Singapore [58]. This may be due to several reasons. In the old city, green plants are cultivated for a long time and grow better. In addition, the streets are narrow. The street density is relatively higher within the buffer distance, leading to a high GVI. On the contrary, in the new development zone of eastern Zhengzhou, the streets are generally wide and sparse. There are simple green belts on both sides of the streets. Thus, the GVI is relatively low (Figure 7). Although most green space in China is public green space provided by the government, from the perspective of SES, the distribution of green space in different communities is still unequal [16,59], which is consistent with the findings of previous studies. The city government and policymakers should give priority to improving the streetscape to ensure that residents in the new development zone can enjoy enough green space.

Based on the LISA cluster map, all types of street view greening correlation (High-High, High-Low, Low-Low, and Low-High) clustered are observed in the interior of the city, while High-Low was observed in the new development zone of eastern Zhengzhou. The Low-High cluster was mainly observed in the central part of the city. The results in this study were inconsistent with previous studies [21,28,30]. This may be the reason that the density of the public transport network in the new development zone of eastern Zhengzhou is not consistent with the growth of street greening. It may also be that the greening strategies implemented in new development zones are not as large as those implemented in central urban areas. In the old city, there is the presence of high vegetation density in downtown streets with appropriate maintenance. These neighborhoods are relatively dilapidated and crowded due to old-fashioned buildings and a high density of population, resulting in relatively low rental prices. This may explain why some neighborhoods within the old city have low rental prices but high green exposure. The High-Low are mainly concentrated in the eastern suburbs of Zhengzhou, which belong to the new development zone of eastern Zhengzhou. The houses are relatively new and the surrounding greening may not be perfect, but there is a large space for development, so the housing price may be relatively high. In addition, previous studies mainly adopted 2D remote sensing data to measure green space around the community [22,29,34]. Due to street-view data limitations inside the community, this study focused on calculating the green space outside the community along the road toward a “15-min city”. The difference in the evaluation methods of green space would lead to inconsistent results.

This study proposes a method to measure the amount of green space based on street view data and machine learning methods. Compared with traditional green space quantitative assessment methods, this method is less time-consuming and more efficient. Thus, there is a significant contribution to explaining the role of green space quality in public health [20].

In the current study, several outputs were highlighted. First, a new framework and perspective were put forward to analyze green space exposure inequity at the community level toward a “15-min city” in Zhengzhou, China. Second, based on street view images, the neighborhood green space was measured in three-dimensional space from a human perspective, then the street view data were combined with machine learning methods to evaluate the street green space environment at a large scale and in a short time. Third, urban rental price big data were used to represent the economic status of the neighborhood, which is more accurate than census data.

This study may also have the following limitations, which need to be further addressed in future studies. First, rental prices were used to reflect residents’ SES without considering relevant factors, such as location, environment, and transportation. In future work, these factors can be incorporated as variables to reflect the SES of residents. Second, the GVI used in this study reflects only the total green proportion of street scenes on both sides of the street, without subdividing green space types. In fact, green spaces with different sizes, comfort, and attractiveness may have different impacts on rental prices, while street view data generally only includes the main road, secondary road, branch road, and other types of roads on which vehicles can drive, collected by a customized camera installed on the automobile roof. Which will lead to a lack of street view pictures of pedestrian streets that vehicles cannot enter. Third, as the exact time when the street view images were taken was not available, the SVIs may not be taken in the same season [58,60]. Fourth, GVI images cover a limited number of observation points and cannot capture the greening of all parts of the city, which may affect the results to some extent. In addition, this paper only focuses on the green exposure of residents along the road, while the green exposure in the community has not been considered due to data limitations. The data can be enriched by collecting green exposure data in the community in the future. A new method is needed for evaluating green space exposure by combining street view data and 2D remote sensing data in the future.

The spatial imbalance of green space exposure in Zhengzhou can provide the basis for municipal and planning departments to further improve the residents’ green space exposure. The spatial mismatch between rental prices and green space exposure helps to implement appropriate greening strategies. Planning efforts should fund green space in Low-Low type neighborhoods around cities to meet the needs of low-income residents for green infrastructure. Authorities and planners should be more sensitive to environmental inequality in green spaces and raise awareness of the problem. Green space cannot be built only in urban centers, affluent neighborhoods, and any major location, because green space has a public character. In addition, the need for green exposure among different social groups should be a key indicator of green planning [61]. Social groups living in disadvantaged communities should be given more green space to ensure the equality of opportunities [7].

## 5. Conclusions

The equitable distribution of green space is increasingly seen as an issue of environmental inequality and has an urgent need to be addressed. Combined with GVI, deep learning, spatial statistical analysis, and the big data of urban rent prices, this study developed and proposed a “15-min city” green space exposure inequity evaluation framework. Instead of remote sensing-based field observations or vegetation indices, we used street view data and deep learning to extract the metrics of green space. Taking Zhengzhou city as an example, this paper explores the relationship between street greening and the social and economic status of residents.

In this study, the location entropy method and bivariate Moran’s I were used to analyze the environmental inequality of residents’ green exposure. The results show that green space exposure inequality is evident among residential communities. The communities in the old city were with relatively high GVI and new city districts were with relatively low GVI. Moreover, a spatially negative correlation was observed between the degree of green space exposure and housing prices. The wealthier communities in the new development zone benefit from low green space, compared with disadvantaged communities in the old city.

In order to realize the “15-min city” and improve the sustainability and livability of urban areas, it is suggested that policymakers and planners pay more attention to the differences in green space exposure among communities. In terms of future planning, we highlight three policy recommendations: (i) the construction of new green spaces should be the priority goal of infrastructure construction in new development zones; (ii) implementing appropriate greening strategies for different types of communities; and (iii) in new development zones, urban development should focus on community-centered walkability.

## Figures and Tables

**Figure 1 ijerph-19-05798-f001:**
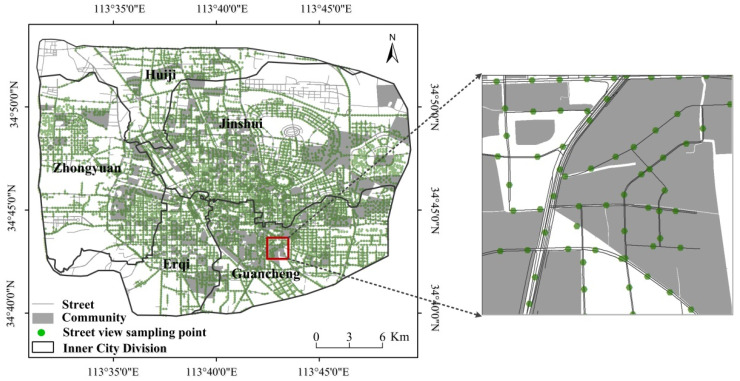
Study area.

**Figure 2 ijerph-19-05798-f002:**
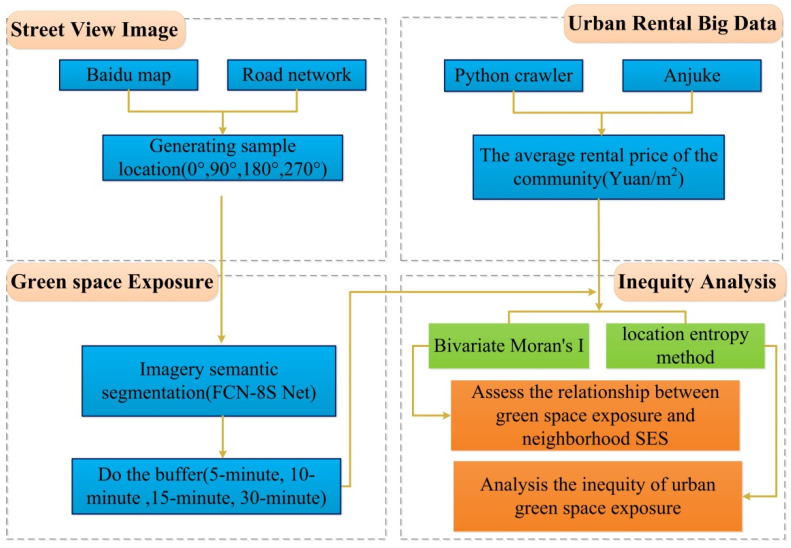
Green space exposure inequity assessment framework. Note: the blue represents input data; the green represents methods; the orange represents outputs.

**Figure 3 ijerph-19-05798-f003:**
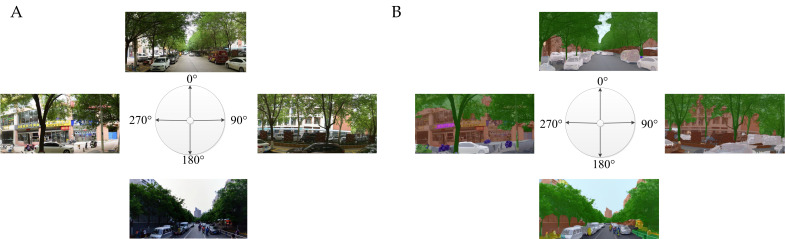
Segmentation results of FCN-8s Net ((**A**) the original images; (**B**) segmentation images).

**Figure 4 ijerph-19-05798-f004:**
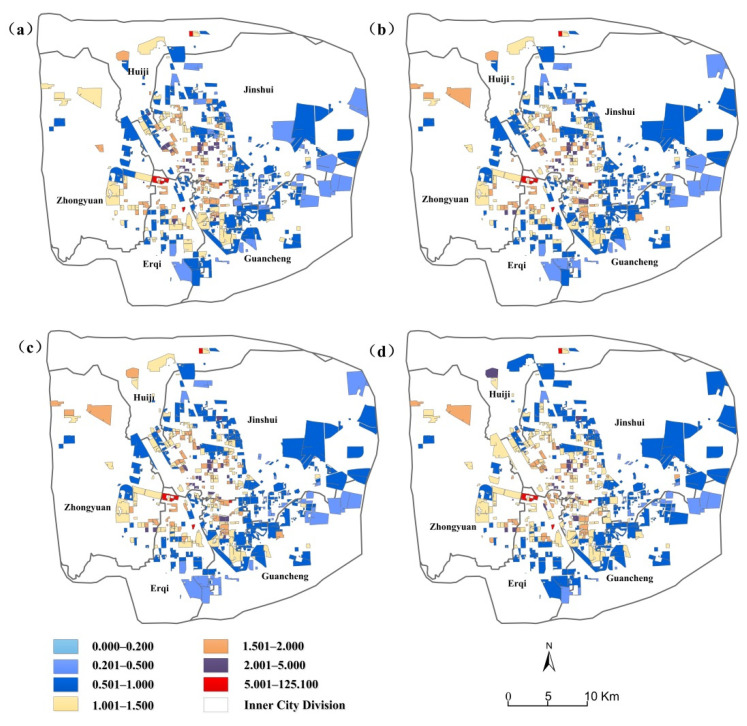
The distribution of the GVI values at the community level ((**a**) GVI of 360 m, (**b**) GVI of 720 m, (**c**) GVI of 1080 m, and (**d**) GVI of 2160 m).

**Figure 5 ijerph-19-05798-f005:**
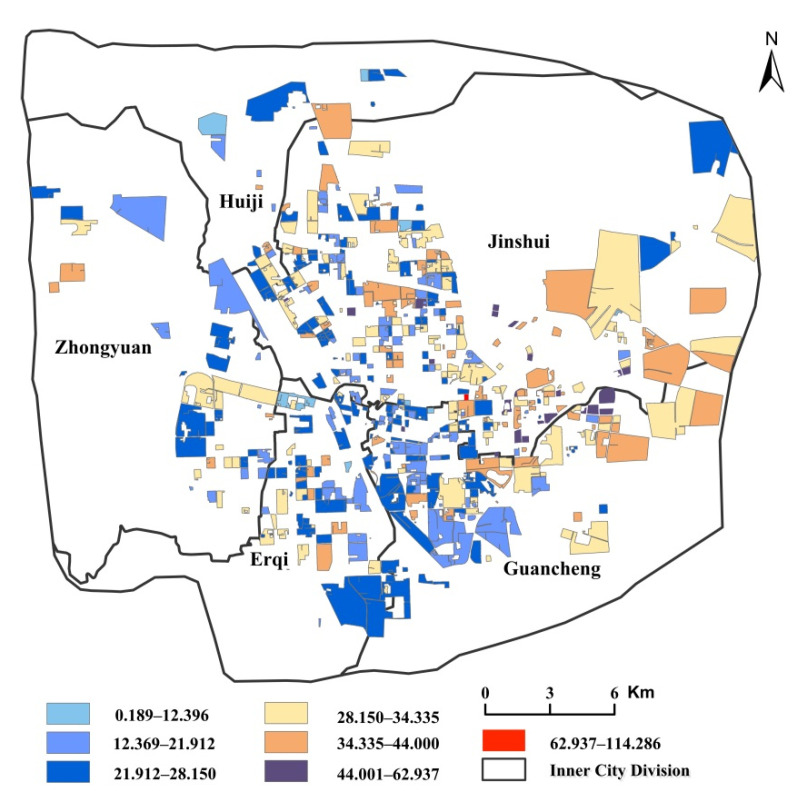
Spatial pattern of rental prices (Yuan/m^2^).

**Figure 6 ijerph-19-05798-f006:**
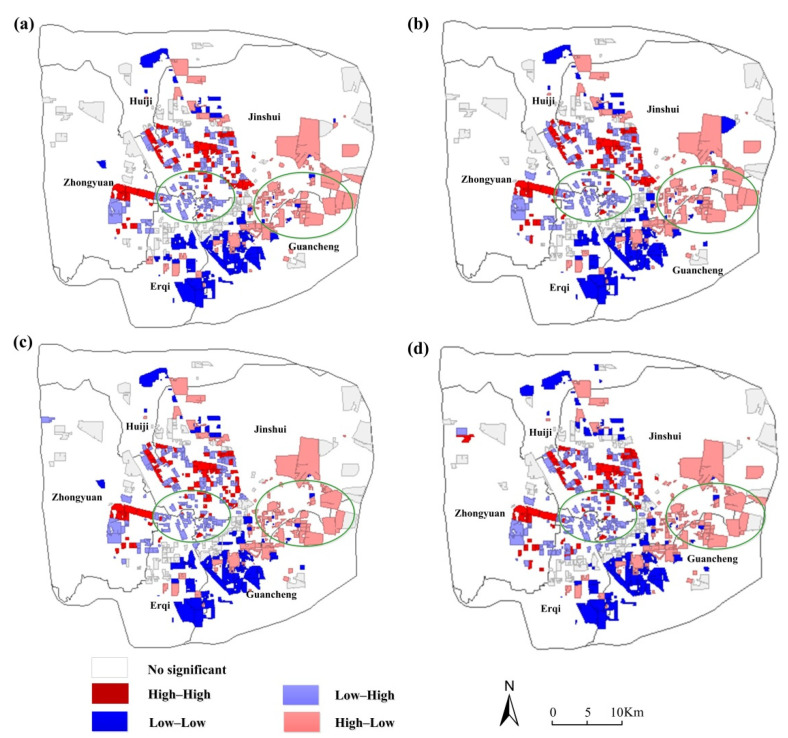
LISA (Local Indicators of Spatial Association) cluster map of the distribution of GVI exposure clusters (The buffer is (**a**) 5 min (360 m), (**b**) 10 min (720 m), (**c**) 15 min (1080 m), and (**d**) 30 min (2160 m) in sequence).

**Figure 7 ijerph-19-05798-f007:**
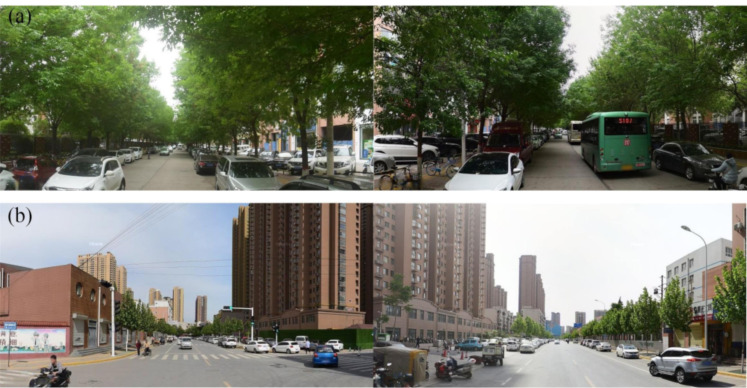
Street view ((**a**) old city; (**b**) new development zone).

**Table 1 ijerph-19-05798-t001:** Statistics of location entropy of each buffer distance (5 min, 10 min, 15 min, and 30 min).

Buffer Distance	LQ	Count (Percentage)
5 min (360 m)	<0.2	3 (0.60)
	0.2–0.5	52 (10.42)
	0.5–1.0	205 (41.08)
	1.0–1.5	122 (24.45)
	1.5–2.0	76 (15.23)
	2.0–5.0	37 (7.42)
	>5.0	4 (0.80)
10 min (720 m)	<0.2	1 (0.20)
	0.2–0.5	41 (8.22)
	0.5–1.0	223 (44.69)
	1.0–1.5	124 (24.85)
	1.5–2.0	69 (13.83)
	2.0–5.0	37 (7.41)
	>5.0	4 (0.80)
15 min (1080 m)	<0.2	0 (0.00)
	0.2–0.5	41 (8.22)
	0.5–1.0	220 (44.09)
	1.0–1.5	135 (27.05)
	1.5–2.0	66 (13.23)
	2.0–5.0	33 (6.61)
	>5.0	4 (0.80)
30 min (2160 m)	<0.2	0 (0.00)
	0.2–0.5	31 (6.21)
	0.5–1.0	204 (40.88)
	1.0–1.5	164 (32.87)
	1.5–2.0	72 (14.43)
	2.0–5.0	24 (4.81)
	>5.0	4 (0.80)

**Table 2 ijerph-19-05798-t002:** Global Moran’s I for the distribution of the GVI values for 5-min, 10-min, 15-min, and 30-min buffer distances.

Buffer Distance	Moran’s I
5 min (360 m)	−0.057 ***
10 min (720 m)	−0.080 ***
15 min (1080 m)	−0.083 ***
30 min (2160 m)	−0.096 ***

*** *p* < 0.001.

## Data Availability

The data and materials used in this study are available from the corresponding author on reasonable request.

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
