# Peer review of "Assessing Inequity in Green Space Exposure toward a “15-Minute City” in Zhengzhou, China: Using Deep Learning and Urban Big Data"

_ijerph, 2022, doi:10.3390/ijerph19105798_

Round 1
Reviewer 1 Report
The presented research paper is highly topical and interesting.
- Please explain the evaluation method: 152-155 This resulted in 7994 sampling points and 31976 street view images. The amount of street view green space for each sampling point was determined as the proportion of the average green space quality of the four images taken from different main directions. WHAT if green is present in a small part of the photo?
- Very interesting research, but conclusions should be developed. It is a pity because many issues were not described in the conclusions.
Author Response
Responses to Comments from Reviewer 1
General Comments
The presented research paper is highly topical and interesting
Response:
We sincerely thank the positive review in this manuscript and for providing valuable comments. We have addressed all issues, and responded point-by-point.
Specific Comments
Comment 1:
Please explain the evaluation method: 152-155 This resulted in 7994 sampling points and 31976 street view images. The amount of street view green space for each sampling point was determined as the proportion of the average green space quality of the four images taken from different main directions. WHAT if green is present in a small part of the photo?
Response:
Thanks for your comment. When this happens, we can locally adjust the interval of sampling points(50m) to obtain the images and correct. Please see Lines 181-182 and Page 6.
Comment 2:
Very interesting research, but conclusions should be developed. It is a pity because many issues were not described in the conclusions.
Response:
Thanks for your comment. In Section 5. Conclusion, we have rewritten the conclusions in detail. The equitable distribution of green space is increasingly seen as an issue of environmental inequality and an urgent need to be addressed. Combined with GVI, deep learning, spatial statistical analysis and big data of urban rent price, this study developed and proposed a ‘15-minute city’ green space exposure inequity evaluation framework. Using green landscape index as the index of street greening, taking Zhengzhou city as an example, this paper explores the relationship between street greening and the social and economic status of residents. In this study, bivariate Moran’s I and location entropy method were used to analyze the environmental inequality of residents' green exposure. The results show that green space exposure inequality is evident among residential communities. The communities in the old city were with relatively high GVI and new city districts were with relatively low GVI. Moreover, spatial uneven association was observed between the degree of green space exposure and housing prices. The wealthier communities in the new development zone benefit low green space, comparing with disadvantaged communities in old city.
These findings are of great significance to the planning and design of street greening in the future. In order to realize the ‘15-minute city’ and improve the sustainability and livability of urban development, it is suggested that policy makers and planners pay more attention to the differences in urban green space exposure. Please see Lines 516-533 and Page 20.
Reviewer 2 Report
This study is a study analyzing the relationship between the green space exposure at the ground level and the socioeconomic status. Appropriate research methods and data are used to achieve the research purpose. However, it seems that improvement is needed in the theoretical review and interpretation of the analysis results.
The authors mention environmental justice at the beginning of the manuscript, and the title of the manuscript is equity assessment. However, the interpretation of the research results does not fit the concept of equity and environmental justice. The authors explain the existence of “low rent-high GVI” regions or “high rent-low GVI” regions as spatial mismatch. It is believed that the authors are assuming that the higher the rent, the higher the GVI should be. In other words, it seems that it is assumed that the neighborhood where people with high SES live should enjoy a high level of green environment. Does this assumption fit the concept of environmental justice or equity? Isn't it a positive phenomenon that there are neighborhood with high GVI among low SES (low rent) neighborhoods? The same question arises in the interpretation process of the location entropy index. Therefore, it is necessary to improve the theoretical review on the concept of environmental justice and spatial equity, and to revise the interpretation of the analysis results from this point of view.
minor comments:
1. SES appears to be an abbreviation for Socioeconomic Status, but it is not mentioned in the manuscript. When SES first appears, please state it clearly.
2. In Table 2, report the p-value of moran's I. And the moran's I values are quite low. Provide a review of previous studies that reveals whether the existence of spatial autocorrelation can be determined with these values.
Author Response
Responses to Comments from Reviewer 2
General Comments
This study is a study analyzing the relationship between the green space exposure at the ground level and the socioeconomic status. Appropriate research methods and data are used to achieve the research purpose. However, it seems that improvement is needed in the theoretical review and interpretation of the analysis results.
The authors mention environmental justice at the beginning of the manuscript, and the title of the manuscript is equity assessment. However, the interpretation of the research results does not fit the concept of equity and environmental justice. The authors explain the existence of “low rent-high GVI” regions or “high rent-low GVI” regions as spatial mismatch. It is believed that the authors are assuming that the higher the rent, the higher the GVI should be. In other words, it seems that it is assumed that the neighborhood where people with high SES live should enjoy a high level of green environment. Does this assumption fit the concept of environmental justice or equity? Isn't it a positive phenomenon that there are neighborhood with high GVI among low SES (low rent) neighborhoods? The same question arises in the interpretation process of the location entropy index. Therefore, it is necessary to improve the theoretical review on the concept of environmental justice and spatial equity, and to revise the interpretation of the analysis results from this point of view.
Response:
Thanks so much for this suggestion. We have carefully reviewed a lot of works related to environmental justice theory. According to Agyeman & Evans (2004), the concept of environmental justice is based on the general principle that all people have a right to be protected from specific environmental issues (e.g., pollution, climate changes) and have access to the same services (greenspaces, social services, transportation). In terms of access to green space, environmental equality refers to the equitable access of residents to green space, regardless of diverse residents’ factors. By contrast, environmental inequality indicates the disproportionate distribution of green space among residents (Nesbitt et al., 2019; Wüstemann et al., 2017).
In this study, environmental equality is analogous to equal opportunity of all communities to obtain green spaces toward ‘15-minute city’ goal, regardless of their socioeconomic status. environmental inequality refers to the unfair distribution of green space.
According to your suggestions, we have carefully improved the theoretical review on the concept of environmental justice and spatial equity in Section1 1. Introduction. Please see Lines 29-41 and Page 2. And to revise the interpretation of the analysis results from this point of view in Section 3. Results. Please see Page 11.
[1] Agyeman, J.; Evans, B. 'just sustainability': the emerging discourse of environmental justice in britain? Geographical Journal. 2004, 170(2), 155–164. https://doi.org/10.1111/j.0016-7398.2004.00117.x.
[2] Nesbitt, L.; Meitner, M.J.; Girling, C.; Sheppard, S.R.J.; Lu, Y. Who has access to urban vegetation? A spatial analysis of distributional green equity in 10 US cities. Landscape and Urban Planning. 2019, 181, 51-79. https://doi.org/10.1016/j.landurbplan.2018.08.007.
[3] Wüstemann, H.; Kalisch, D.; & Kolbe, J.. Access to urban green space and environmental inequalities in Germany. Landscape and Urban Planning. 2017, 164, 124–131. https://doi.org/10.1016/j.landurbplan.2017.04.002.
Specific Comments
Comment 1:
SES appears to be an abbreviation for Socioeconomic Status, but it is not mentioned in the manuscript. When SES first appears, please state it clearly.
Response:
Thanks for your insightful comment. We have modified the SES as socioeconomic status (SES), when this term first appears. Please see Lines 67-68 and Page 3.These studies reported that urban green exposure varies according to socioeconomic status (SES).
Comment 2:
In Table 2, report the p-value of moran's I. And the moran's I values are quite low. Provide a review of previous studies that reveals whether the existence of spatial autocorrelation can be determined with these values.
Response:
Thanks for your comment. We have carefully checked our data and made some modification. In Section 3.2 The association between green space exposure and housing prices, we have rewritten results about global bivariate Moran’s I.
Table 2 . Global Moran’s I for distribution of the GVI values for 5-minute, 10-minute, 15-minute and 30-minute buffer distance.
|
Buffer distance |
Moran's I |
|
5-minute (360 m) |
-0.057*** |
|
10-minute (720m) |
-0.080*** |
|
15-minute (1080m) |
-0.083*** |
|
30-minute (2160m) |
-0.096*** |
*p < 0.05, **p < 0.01, ***p < 0.001.
In this study, first, we adopted the global bivariate Moran’s I to measure the correlation between the green space exposure and socioeconomic status. The result of global bivariate Moran’s I are -0.057(360m), -0.080(720m), -0.083(1080m) and -0.096(2160m) respectively, revealing a negative spatial correlation between housing price and green space exposure. That is, a community with a higher housing price generally enjoys low green space exposure. Next, the bivariate LISA was applied for locally examining the association between socioeconomic conditions and green space exposure. Please see Lines 333-339 and Page13.
We have carefully reviewed previous studies related to the method of global bivariate Moran’s I. Previous studies showed that the relative low values of global bivariate Moran’s I illustrated the existence of a statically significant association between two variables (Liu et al.,2020; Moraes et al.,2018).
[1] Liu, Y.; Zhang, X.; Pan, X..; Ma, X.; Tang, M. The spatial integration and coordinated industrial development of urban agglomerations in the Yangtze River Economic Belt, China. Cities. 2020, 104,102801. https://doi.org/10.1016/j.cities.2020.102801.
[2] Arcoverde, M.A.M.; Berra, T.Z.; Alves, L.S.; Santos, D.T.D.; Belchior, A.S.; Ramos, A.C.V.; Assis, l.S.d.; Neto, F.C.; Silva-Sobrinho, R.A.; Nihei, O.K.; Arcencio, R.A. How do social-economic differences in urban areas affect tuberculosis mortality in a city in the tri-border region of Brazil, Paraguay and Argentina. BMC Public Health. 2018, 18(1),795-809. https://doi.org/10.1186/s12889-018-5623-2.
Reviewer 3 Report
The scope of the article is clearly articulated and well presented by adequately foregrounding its topicality, urgency and strategic role in supporting policy enhancements in the field of spatial justice concerning green urban infrastructure and people’s wellbeing. Its conceptual approach is sound and underpinned by a cogent discussion of the limits of the current analytical methods. However, the article shows misalignments between identifying and articulating the problem and designing the methodology designed to address it. This shortcoming mainly regards focusing on one aspect of the green infrastructure: streetscape perception. Such limit leads to a gross overinterpretation of findings and must be revised. It is also worth noting that the revision of the article should address (i) the reason for using data taken from vehicular lanes of streets to represent the pedestrian experience, (ii) the seasonal aspects regarding the collected images, (iii) the uncommon extension of pedestrian catchments (up to 30 minutes), (iv) the exclusion of other transportation means (e.g., bicycle and PT), (v) the reason why, notwithstanding the very well-established methods to calculate ped-sheds, the research uses sheer circular surfaces instead of areas determined by connectivity analysis.
Author Response
Responses to Comments from Reviewer 3
General Comments
The scope of the article is clearly articulated and well presented by adequately foregrounding its topicality, urgency and strategic role in supporting policy enhancements in the field of spatial justice concerning green urban infrastructure and people’s wellbeing. Its conceptual approach is sound and underpinned by a cogent discussion of the limits of the current analytical methods. However, the article shows misalignments between identifying and articulating the problem and designing the methodology designed to address it. This shortcoming mainly regards focusing on one aspect of the green infrastructure: streetscape perception. Such limit leads to a gross overinterpretation of findings and must be revised.
Response:
Thanks for your comment. We have tried our best to revise the manuscript according to your construction comments and suggestions.
Major comments:
Comment 1:
the reason for using data taken from vehicular lanes of streets to represent the pedestrian experience.
Response:
Thanks for your comment. In 2018, the new urban Residential Area Planning and Design Standard released by the Ministry of Housing and Urban-Rural Development, the "15-minute city ", "10-minute city" and "5-minute city" are taken as the core objects of residential area planning and facility configuration. In the "Technical Guide for Planning community Living Circle", it is proposed to configure various functions and facilities (elderly services, medical care, education, commerce, transportation, sports, etc.) for residents' basic life by taking the 15-minute walking distance as the space scale. In addition, the Green Vision Index (GVI) was developed to assess the visibility of the surrounding urban forest using color images as a proxy for pedestrians' perception of green space (Yang et al., 2009). Recent developments in machine learning methods combined with online map data allow people to combine the sentiment of residents towards green Spaces from social media (Flickr) with ground objects (e.g., trees and grass) from interactive panormas (e.g.,Street View Images(SVI)) to better capture indicators of the quality of urban green Spaces (Helbich et al., 2019; Li et al., 2018). Therefore, in this study, we adopted street view images data and GVI index to analysis of the green space exposure patterns based on the ground-based view. Please see Lines 94-103 and Pages 3-4.
[1] Yang, J.; Zhao, L.; McBride, J.; & Gong, P. Can you see green? Assessing the visibility of urban forests in cities. Landscape and Urban Planning. 2009, 91(2), 97-104. https://doi.org/10.1016/j.landurbplan.2008.12.004.
[2] Helbich, M.; Yao, Y.; Liu, Y.; Zhang, J.; Liu, P.; Wang, R. Using deep learning to examine street view green and blue spaces and their associations with geriatric depression in Beijing, China. Environ Int. 2019,126, 107-117. https://doi.org/10.1016/j.envint.2019.02.013.
[3] Li, X.; Santi, P.; Courtney, T.K.; Verma, S.K.; Ratti, C. Investigating the association between streetscapes and human walking activities using Google Street View and human trajectory data. Transactions in GIS. 2018, 22(4), 1029-1044. https://doi.org/10.1111/tgis.12472.
Comment 2:
the seasonal aspects regarding the collected images?
Response:
Thanks for your comment. In Section 2.2.1 Street view images data, we have added some sentences to introduce the seasonal aspects regarding the collected images. “In this study, street view images are collected in 2020, with no specific capture-time information. Some images were obtained during non-green seasons. We visually checked the vegetation conditions in each image and deleted those sites with images captured during non-green seasons. we locally adjusted the interval of sampling points(50m) to obtain more images and correct.” Please see Lines 178-182 and Page6.
In addition, in Section 4. Discussion, we listed the data limitation. “Third, the exact time when the street View image was taken was not available and SVI may not be taken in the same season (Ye et al., 2018; Nagata et al., 2020).” Please see Lines 499-500 and Page 19.
[1] Ye, Y.; Richards, D.; Lu, Y.; Song, X.; Zhuang, Y.; Zeng, W.; Zhong, T. Measuring daily accessed street greenery: A human-scale approach for informing better urban planning practices. Landscape and Urban Planning. 2018, 191. https://doi.org/10.1016/j.landurbplan.2018.08.028.
[2] Nagata, S.; Nakaya, T.; Hanibuchi, T.; Amagasa, S.; Kikuchi, H.; Inoue, S. Objective scoring of streetscape walkability related to leisure walking: Statistical modeling approach with semantic segmentation of Google Street View images. Health Place. 2020, 66, 102428. https://doi.org/10.1016/j.healthplace.2020.102428.
Comment 3:
the uncommon extension of pedestrian catchments (up to 30 minutes)?
Response:
Thanks for your suggestion. According to Li et al. (2008) and Chen et al. (2021), we found that the distance factor was the main factor affecting residents' travel. For the purpose of walking and exercising, the maximum psychological endurance time of travelers was usually 30 minutes Therefore, 30min is selected as the maximum buffer radius in this study. Please see Lines 185-189 and Page 7.
[1] Li, b.; Song, Yun.; Yu Kong.jian. Evaluation Method for Measurement of Accessibility in Urban Public Green Space Planning. Acta Scientiarum Naturalium Universitatis Pekinensis, 2008 ,44(4), 618-624. https://doi.org/10.13209/j.0479-8023.2008.096.
[2] Chen, Y.; Yu, P.; Li, Z.; Wang, J.; Chen, Y. Environment Equity Measurement of Urban Green Space from the Perspective of SDG11: A Case Study of the Central Urban Area of Wuhan. Geography and Geo-Information Science. 2021, 37(04), 81-89. https://doi.org/10/3969/j.issn.167-0504.2021.04.012.
Comment 4:
the exclusion of other transportation means (e.g., bicycle and PT)
Response:
Thanks for your comment. The essence of "15-minute City " is to create a community environment where most of the public service facilities (elderly services, medical care, education, commerce, transportation, sports, etc.) can be reached within a 15-minute walking distance. Therefore, in this study, we only consider the walking transportation means.
[1] Standard for urban residential area planning and design GB50180-2018. https://www.soujianzhu.cn/NormAndRules/NormContent.aspx?id=362&msclkid=2838fc4db6e611ecb2f8732be123e32d.
Comment 5:
the reason why, notwithstanding the very well-established methods to calculate ped-sheds, the research uses sheer circular surfaces instead of areas determined by connectivity analysis.
Response:
Thanks for your comment. The boundary shapes of each community are different, and most of them are irregular. In order to accurately evaluate the green space exposure of each community within the 15-minute walking range, this study adopts the boundary of the community as the benchmark to establish buffer and measure the green space exposure, no considering the neighborhood as a point and building buffers based on road network.
Round 2
Reviewer 2 Report
It seems that the revisions have been appropriately reflected in the points of the first review.
Author Response
Responses to Comments from Reviewer 2
General Comments
It seems that the revisions have been appropriately reflected in the points of the first review.
Response:
Thanks for your comments. In the revised manuscript, we have added some contents in literature review and discussion respectively and try to illustrate the research problem and deficiencies of this study.
(1)Introduction Section. We summarized the measurement methods and data of green space in previous studies, pointed out their shortcomings. Please see Lines 87-107 and Pages 3-4. We put forward the aim of this study. Please see Lines 150-161 and Page 5.
(2)Discussion Section. We have added some sentences to explain the deficiencies of this study. “In addition, this study only focuses on the green exposure of residents along the road, while the green exposure in the community has not been considered due to the data limitation. The data can be enriched by collecting the green exposure data in the community in the future. And the new method is needed for evaluating green space exposure combining street view data and 2D remote sensing data in the future.” Please see Lines 535-540 and Pages 20-21.

Reviewer 3 Report
The revised manuscript is a polished version of the original submission and does not address the issues highlighted in the review.
Author Response
Responses to Comments from Reviewer 3
General Comments
The revised manuscript is a polished version of the original submission and does not address the issues highlighted in the review.
Response:
Thanks for your comments. We have read your comments carefully and tried our best to revise the manuscript according to your comments and suggestions. We have focused on the following questions.
Comment 1:
The scope of the article is clearly articulated and well presented by adequately foregrounding its topicality, urgency and strategic role in supporting policy enhancements in the field of spatial justice concerning green urban infrastructure and people’s wellbeing. Its conceptual approach is sound and underpinned by a cogent discussion of the limits of the current analytical methods. However, the article shows misalignments between identifying and articulating the problem and designing the methodology designed to address it. This shortcoming mainly regards focusing on one aspect of the green infrastructure: streetscape perception. Such limit leads to a gross overinterpretation of findings and must be revised.
Response:
Thanks for your comments. We have tried our best to revise the manuscript according to your construction comments and suggestions. According to your suggestions, we have added some contents in literature review, results analysis and discussion respectively, and try to explain the research problem, method design, results analysis and disadvantages in this study.
(1)Introduction Section. We summarized the measurement methods and data of green space in previous studies, pointed out their shortcomings. Please see Lines 87-107 and Pages 3-4.We put forward the aim of this study. Please see Lines 150-161 and Page 5.
(2)Data source Section. We pointed out the concept and measurement methods of green space. “In this study, the green space refers to areas of grass and trees in the image, as measured by the GVI. The GVI is the ratio of the total green area of four images taken at an intersection to the total area of the four images.”Please see Lines 190-192 and Page 7.
(3)Result Section. The locational entropy of less than 0.5, 1 and greater than 2 were analyzed previously. In order to avoid overinterpretation of findings, we retained the classification results with large difference between location entropy and 1, deleted sentences of entropy greater than 0.5 and less than 2. We only focused on the results that the locational entropy is less than 0.5 and greater than 2. Please see Lines 329-359 and Pages 12-13.
(4)Discussion Section. We have added some sentences to explain the deficiencies of this study. “In addition, this study only focuses on the green exposure of residents along the road, while the green exposure in the community has not been considered due to the data limitation. The data can be enriched by collecting the green exposure data in the community in the future. And the new method is needed for evaluating green space exposure combining street view data and 2D remote sensing data in the future.” Please see Lines 535-540 and Pages 20-21.
[1] You, H. Characterizing the inequalities in urban public green space provision in Shenzhen, China. Habitat International. 2016, 56, 176-180. http://dx.doi.org/10.1016/j.habitatint.2016.05.006.
[2] Wang, R.; Feng, Z.; Pearce, J.; Yao, Y.; Li, X.; Liu, Y. The distribution of greenspace quantity and quality and their association with neighbourhood socioeconomic conditions in Guangzhou, China: A new approach using deep learning method and street view images.
[3] Xu, C.; Haase, D.; Pribadi, DO.; Pauleit, S. Spatial variation of green space equity and its relation with urban dynamics: A case study in the region of Munich. Ecological Indicators. 2018, 93, 512-523. https://doi.org/10.1016/j.ecolind.2018.05.024.
[4] Apparicio, P.; Thi-Thanh-Hien, Pham.; Seguin, A.M.; Landry, S.; Gagnon, M. Spatial distribution of vegetation in Montreal: An uneven distribution or environmental inequity? Landscape and Urban Planning. 2012, 107(3), 214-224. http://dx.doi.org/10.1016/j.landurbplan.2012.06.002.
[5] A, Rigolon.; M.H.E.M, Browning .; K, Lee.; S, Shin. Access to urban green space in cities of the Global South: A systematic literature review. Urban Science.2018, 2 (3), pp. 67-91. https://doi.org/10.3390/urbansci2030067.
[6] Wüstemann, H.; Kalisch, D.; Kolbe, J. Access to urban green space and environmental inequalities in Germany. Landscape Urban Plann. 2017,164, 124–131. https://doi.org/10.1016/j.landurbplan.2017.04.002.
[7] Yao, L.; Liu, J.; Wang, R..; Yin, K.; Han, B. Effective green equivalent—A measure of public green spaces for cities. Ecological Indicators. 2014, 47, 123–127. https://doi.org/10.1016/j.ecolind.2014.07.009.
[8] Jensen, R.; Gatrell, J.; Boulton, J.; Harper, B. Using remote sensing and geographic information systems to study urban quality of life and urban forest amenities. Ecol. Soc. 2004, 9 (5), 5. https://sci-hub.yncjkj.com/10.1016/j.ecolecon.2004.07.005.
[9] Jennings, V.; Johnson Gaither, C.; Gragg, R.S. Promoting Environmental Justice Through Urban Green Space Access: A Synopsis. Environmental Justic.2012, 5(1), 1-7. https://doi.org/10.1089/env.2011.0007.
[10] Leslie, E.; Sugiyama, T.; Ierodiaconou, D.; Kremer, P. Perceived and objectively measured greenness of neighbourhoods: Are they measuring the same thing? Landscape and Urban Planning. 2010, 95(1-2), 28-33. http://dx.doi.org/10.1016/j.landurbplan.2009.11.002.
[11] Landry, S.M.; Chakraborty, J. Street Trees and Equity: Evaluating the Spatial Distribution of an Urban Amenity. Environment and Planning A: Economy and Space. 2009, 41(11), 2651-2670. https://doi.org/10.1068/a41236.
Comment 2:
the reason why, notwithstanding the very well-established methods to calculate ped-sheds, the research uses sheer circular surfaces instead of areas determined by connectivity analysis.
Response:
Thanks for your comment. The boundary shapes of each community are different, and most of them are irregular. In order to accurately evaluate the green space exposure of each community within the 15-minute walking range, this study adopts the boundary of the community as the benchmark to establish buffer and measure the green space exposure.
According to your suggestions, we have constructed the road network and calculated areas determined by connectivity analysis. The results are similar with our previous results that obtained by using sheer circular surfaces.
(1) Spatial inequity of green space exposure
The location entropy of the community greater than 1 indicates that the urban green space services enjoyed by the population with unit income in the region are higher than the overall average level. While the location entropy of the community less than 1 indicates that the urban green space services enjoyed by the population with unit income in the region are lower than the overall average level.
Table 1. Statistical of location entropy of each buffer distance (5-minute,10-minute,15-minute and 30-minute).
|
Buffer distance |
LQ |
Count (percentage) |
|
5-minute (360 m) |
<0.2 |
31 (6.21) |
|
0.2-0.5 |
85 (17.03) |
|
|
0.5-1.0 |
160 (32.06) |
|
|
1.0-1.5 |
104 (20.84) |
|
|
1.5-2.0 |
52 (10.43) |
|
|
2.0-5.0 |
64 (12.83) |
|
|
>5.0 |
3 (0.60) |
|
|
10-minute (720 m) |
<0.2 |
8 (1.60) |
|
0.2-0.5 |
60 (12.02) |
|
|
0.5-1.0 |
199 (39.88) |
|
|
1.0-1.5 |
116 (23.25) |
|
|
1.5-2.0 |
67 (13.43) |
|
|
2.0-5.0 |
45 (9.02) |
|
|
>5.0 |
4 (0.80) |
|
|
15-minute (1080 m) |
<0.2 |
1 (0.20) |
|
0.2-0.5 |
44 (8.82) |
|
|
0.5-1.0 |
209 (41.88) |
|
|
1.0-1.5 |
142 (28.46) |
|
|
1.5-2.0 |
61 (122.22) |
|
|
2.0-5.0 |
38 (7.62) |
|
|
>5.0 |
4 (0.80) |
|
|
30-minute (2160 m) |
<0.2 |
1 (0.20) |
|
0.2-0.5 |
42 (8.42) |
|
|
0.5-1.0 |
202 (40.48) |
|
|
1.0-1.5 |
159(31.86) |
|
|
1.5-2.0 |
61 (12.23) |
|
|
2.0-5.0 |
30 (6.01) |
|
|
>5.0 |
4 (0.80) |
Table 1 shows the location entropy value of each spatial unit within the buffer zone of the residential area, which is divided into 7 levels for a certain income level and the average occupancy of urban public green space. Figure 1 shows the spatial distribution pattern of location entropy values of each community. In order to analyze the unfairness of urban green space more accurately, we only focused on the results that the locational entropy is less than 0.5 and greater than 2. First of all, the areas with very low location entropy (the location entropy value lower than 0.5, that is, the urban public green space service enjoyed by each income level is less than half of the average level) contain 116 in the buffer zone of 360m, accounting for 23.24% of the total number of cells, and 68 in the buffer zone of 720m, accounting for 13.62% of the total number of cells, and 45 in the buffer zone of 1080m, accounting for 9.02% of the total number of cells. Moreover, there are 43 cells within the 2160m buffer zone, accounting for 8.62% of the total number of cells. The second, areas with high location entropy (location entropy value is higher than 2, that is, the urban public green space service enjoyed by each income level is twice higher than the average level), in the 360m buffer contains 67, accounting for 13.43% of the total cell number, in the 720m buffer contains 49, accounting for 9.82% of the total cell number, in the 1080m buffer contains 42, accounting for 8.42% of the total cell number, in the 2160m buffer contains 34, accounting for 6.81% of total cell number.
As seen in Figure 1, the location entropy index showed a gradual decrease in the circle structure, the number of communities with higher locational entropy index decreases from the inner ring to the periphery of the city, forming a circle layer structure. They can be highlighted that there is a inequality mismatch in the allocation of green public space resources in the residential areas in the study area. The communities with lower locational entropy index are mainly distributed along the outer ring. The communities with lower location entropy index were mainly distributed in the periphery of the city, including the east of Jinshui District, the east of Erqi District, and the of northeast Guancheng Minority District. The communities with higher location entropy index were mainly distributed in the city center, including the west of Jinshui District, the east of Zhongyuan District, the north of Erqi District, and the northwest of Guancheng Minority District.
Figure 1. The distribution of the GVI values at the community level ( (a)is GVI of 360m, (b) is GVI of 720m, (c)is GVI of 1080m and (d)is GVI of 2160m)
(2) The association between green space exposure and rental prices
This study takes the residential area as the center, makes buffer polygons with road network distances of 360m, 720m, 1080m and 2160m, and calculates the exposed area of green space within the buffer polygon. Finally, the correlation analysis between the exposed area of green space in the buffer polygon and the per capita income of the community is made. These correlation coefficients were used to access the relationship between green space and SES. As can be seen from Table 2, rent price is negatively correlated with GVI on the whole.
Table 2 . Global Moran’s I for distribution of the GVI values for 5-minute, 10-minute, 15-minute and 30-minute buffer distance.
|
Buffer distance |
Moran's I |
|
5-minute (360 m) |
-0.025** |
|
10-minute (720m) |
-0.055*** |
|
15-minute (1080m) |
-0.078*** |
|
30-minute (2160m) |
-0.088*** |
*p < 0.05, **p < 0.01, ***p < 0.001.
As shown in Figure 2, local Moran's I revealed uneven spatial distribution between green space and different income groups, especially in High-Low (high housing price and low GVI) types and Low-High (low housing price and high GVI).
As shown in Figure 2a, High–Low type (high housing price and low GVI), comprised of 71 residential areas are mainly distributed along the east New city district. Low-High type (low housing price and high GVI) are comprised of 107 residential areas concentrated in north part of Guancheng Minority and the southwest part of Jinshui District. On the other hand, Low-Low type (low housing price and low GVI) comprised of 75 residential areas located in the northeastern part of Erqi District and western part of Guancheng Minority District. While High-High type (high housing price and high GVI) consists of 58 residential areas, mainly concentrated in the western part of Jinshui District.
As shown in Figure 2b, regarding the 720m buffer zone considered, the results showed that High-Low type (high housing price and low GVI) comprised of 103 communities, mainly concentrated in the north part of Guancheng Minority and the east New city district, while Low-High type (low housing price and high GVI) comprised of 124 residential areas, mainly distributed the southwest part of Jinshui District. Moreover, Low-Low type (low housing price and low GVI) comprised of 89 residential areas, mainly concentrated in the northeast of Erqi District and the western part of Guancheng Minority District, whereas High-High type (high housing price and high GVI) comprised of 72 residential areas, mainly concentrated in the western part of Jinshui District.
As shown in Figure 2c, regarding the 1080m buffer zone considered, the High-Low type (high housing price and low GVI) consists of 108 communities located in the north part of Guancheng Minority and the east New city district. While Low-High type (low housing price and high GVI) consists of 128 residential areas, mainly distributed in the Zhongyuan district and the western part of Jinshui District. Additionally, Low-Low type (low housing price and low GVI) consists of 80 residential areas, mainly concentrated in the east of Erqi District and the western part of Guancheng Minority District, whereas High-High type (high housing price and high GVI) consists of 73 residential areas, mainly concentrated in the western part of Jinshui District.
On the other hand, regarding the 2160m(Figure 2d) buffer zone considered, the High-Low type (high housing price and low GVI) consists of 105 communities distributed in the north part of Guancheng Minority . While Low-High type (low housing price and high GVI) consists of 135 residential areas , mainly distributed in the Zhongyuan district, the western part of Jinshui District and the north part of Erqi District .Moreover, Low-Low type (low housing price and low GVI) comprised of 83 residential areas, mainly concentrated in the east of Erqi District and the western part of Guancheng Minority District, whereas High-High type (high housing price and high GVI) comprised of 77 residential areas, mainly concentrated in Jinshui District.
As shown in Figure 2, Low-High type (low housing price and high GVI) residential areas are mainly within the inner ring road, including the old urban area. This may be due to the shabby houses and traffic congestion in the old city, and the housing price will be relatively low. However, the old urban area usually with high street density and good greening, is more suitable for residents to walk leisure. The GVI will be relatively high. High-Low type (high housing price and low GVI) are mainly distributed along the east New city district. This may be because the eastern part of Zhengzhou is a new development zone with large development space, so the housing price is relatively high. However, various facilities are not perfect and the green environment is under planning, so the GVI value is relatively low.
Figure 2. LISA (Local Indicators of Spatial Association) cluster map of distribution of GVI exposure clusters (The buffer is (a) 5-minute (360 m), (b) 10-minute (720 m), (c) 15-minute (1080 m) and (d) 30-minute (2160 m) in sequence)
